# Biodegradation and Metabolic Pathway of 17β-Estradiol by *Rhodococcus* sp. ED55

**DOI:** 10.3390/ijms23116181

**Published:** 2022-05-31

**Authors:** Irina S. Moreira, Sapia Murgolo, Giuseppe Mascolo, Paula M. L. Castro

**Affiliations:** 1CBQF—Centro de Biotecnologia e Química Fina, Laboratório Associado, Escola Superior de Biotecnologia, Universidade Católica Portuguesa, Rua Diogo Botelho 1327, 4169-005 Porto, Portugal; plcastro@ucp.pt; 2CNR, Istituto di Ricerca Sulle Acque, Via F. De Blasio 5, 70132 Bari, Italy; sapia.murgolo@ba.irsa.cnr.it (S.M.); giuseppe.mascolo@ba.irsa.cnr.it (G.M.); 3CNR, Istituto di Ricerca per la Protezione Idrogeologica, Via Amendola 122 I, 70126 Bari, Italy

**Keywords:** endocrine disrupting chemicals, *Rhodococcus* sp. ED55, bioaugmentation, wastewater, 17β-estradiol

## Abstract

Endocrine disrupting compounds (EDCs) in the environment are considered a motif of concern, due to the widespread occurrence and potential adverse ecological and human health effects. The natural estrogen, 17β-estradiol (E2), is frequently detected in receiving water bodies after not being efficiently removed in conventional wastewater treatment plants (WWTPs), promoting a negative impact for both the aquatic ecosystem and human health. In this study, the biodegradation of E2 by *Rhodococcus* sp. ED55, a bacterial strain isolated from sediments of a discharge point of WWTP in Coloane, Macau, was investigated. *Rhodococcus* sp. ED55 was able to completely degrade 5 mg/L of E2 in 4 h in a synthetic medium. A similar degradation pattern was observed when the bacterial strain was used in wastewater collected from a WWTP, where a significant improvement in the degradation of the compound occurred. The detection and identification of 17 metabolites was achieved by means of UPLC/ESI/HRMS, which proposed a degradation pathway of E2. The acute test with luminescent marine bacterium *Aliivibrio fischeri* revealed the elimination of the toxicity of the treated effluent and the standardized yeast estrogenic (S-YES) assay with the recombinant strain of *Saccharomyces cerevisiae* revealed a decrease in the estrogenic activity of wastewater samples after biodegradation.

## 1. Introduction

Environmental contamination with estrogens is considered problematic for wildlife and human health. Natural estrogens, such as estrone (E1), 17β-estradiol (E2) and estriol (E3), are reproductive steroidal hormones produced by all vertebrates and some invertebrate’s organisms. Additionally, synthetic estrogens, such as 17α-ethinylestradiol (EE2) and 3-methyl ether of ethinylestradiol (MeEE2), are used in contraceptive pills and in other drugs administered during menopause. Steroidal hormones excreted in urine and feces are considered the main source of endocrine disruptors to the environment. Conventional wastewater treatment plants (WWTPs) present varying efficiency degrees for the removal of these compounds; despite being incomplete, biodegradation is considered the main removal mechanism [1]. Moreover, during treatment, deconjugation of lower estrogenic sulfate, glucuronide or sulfoglucuronide forms, results in increased estrogenic activity of the wastewater effluents [2]. Another source of contamination is livestock wastes [3,4]. Besides the natural and synthetic hormones, another important contaminant is the xenoestrogen bisphenol-A (BPA), one of the most abundant industrial synthetic chemicals produced globally, used to manufacture consumer products, such as water bottles, water pipes, and food cans [5]. BPA presents estrogenic activity, among other endocrine-disrupting effects, and is present ubiquitously in the environment due to its widespread use [6].

Consequently, estrogens and others endocrine disruptors chemicals (EDCs) are continuously released into the environment via effluent discharge, resulting in their occurrence in the range of low microgram per liter to few nanogram per liter levels in rivers [7,8,9], lakes [10], sediments [8,11] and even in drinking water reservoirs [12]. Due to the high concern posed by these contaminants, E2 and EE2 were added to the first EU watch list of emerging substances to be monitored in 2013 (Directive 2013/39/EU). Concentrations of E2 and EE2 in surface water are typically below 50 ng/L and 10 ng/L, respectively [1], but these values greatly exceed the European Union proposed annual average environmental quality standard (0.04 and 0.035 ng/L, respectively) [1]. When present in the environment, they can cause endocrine disruption and significant adverse effects on the physiological function and reproductive system in humans and animals [13,14].

Biodegradation is considered an interesting strategy for the removal of EDCs because of its low cost, reduced environmental impact and efficiency in the removal of dispersed organics, even at low concentrations. Microorganisms with the ability to biodegrade or biotransform E2 have been isolated from different environmental compartments. Five bacterial strains of *Rhodococcus* sp. and *Sphingomonas* sp. capable of utilizing E2 were isolated from soil samples [15]. Fernández and coworkers isolated five bacterial strains with the ability to degrade E2 from deep sea sediments [16]. The E2 degrading strain *Stenotrophomonas maltophilia* SJTL3 was isolated from wastewater [17]. Various E2-degrading strains were isolated from activated sludge of wastewater treatment plants, namely five bacterial strains belonging to the genus *Bacillus* [18], *Acinetobacter* sp. strain [19], *Novosphingobium* sp. E2S [20], *Rhodococcus* sp. JX-2 [21], *Rhodococcus* sp. DS201 isolated from a municipal wastewater treatment plant that receives effluents from a contraceptive factory in Beijing, China [22]. Yu and coworkers also isolated 14 phylogenetically diverse bacteria for the degradation of E2 from activated sludge [23]. In most of the studies, the biodegradation was only evaluated in synthetic medium and not in real wastewater and there is no evaluation of the toxicity and estrogenic activity of the final products of biodegradation.

The main objective of the present study was to investigate the biodegradation of E2 by a bacterial strain isolated from sediments of a discharge point of a municipal wastewater treatment plant in Coloane, Macau. The potential for bioaugmentation with this strain for the improvement of E2 removal in real wastewater and for reducing toxicity and estrogenic activity were evaluated. Metabolites produced during E2 degradation were identified through ultra-performance liquid chromatography coupled with high-resolution mass spectrometry (UPLC-ESI-HRMS) and used to propose a metabolic pathway of degradation.

## 2. Results and Discussion

### 2.1. Phylogenetic Analysis of E2 Degrading Strain

The E2 degrading strain ED55 was previously isolated from sediments of a discharge point of a WWTP in Coloane (Macau) for their capacity to degrade EDCs, when supplied as a sole source of carbon [24]. Strain ED55 morphology was examined by light microscopy, which revealed rod-shaped cells. The strain was positive to the Gram staining method. When streaked onto nutrient agar (NA) plates, opaque colonies with a pink color and regular edges were formed. Phylogenetic analysis based on 16S rRNA gene sequences showed that strain ED55 is a member of the genus *Rhodococcus* and formed a consistent cluster with *Rhodococcus ruber* (Acc. No. 118602.1; similarity = 99.9%) and *Rhodococcus aetherivorans* (Acc. No. 118619.1; similarity = 99.4%) (Figure 1).

### 2.2. Biodegradation of E2 in Mineral Salts Medium

The degradation of E2 as a sole carbon source by *Rhodococcus* sp. ED55 was evaluated at 5 mg/L. The data in Figure 2 reveal that strain ED55 was able to completely degrade the supplied amount of E2 in 4 h. The rate of degradation obtained was 0.659 ± 0.050 h^−1^ (R^2^ = 0.96) and a half-life of 1.06 ± 0.08 h. The addition of acetate as a supplementary carbon source did not affect E2 degradation. In the non-inoculated control assays and in the control inoculated with autoclaved *Rhodoccocus* sp. ED55, a decrease in E2 concentration was not observed, indicating that neither abiotic degradation nor adsorption occurred under the tested conditions.

The bacterial strain *Rhodococcus* sp. ED55 was able to degrade E2 at much faster rates than those reported in previous studies, even for other *Rhodococcus* strain. Most biodegradation studies available in the literature were also carried out in batch assays using synthetic mineral salts media (MM). *Rhodococcus* sp. DS201 isolated from the activated sludge of a municipal treatment plant degraded 1 mg/L of E2 in 3 days at 30 °C, with an initial pH of 7 and an inoculum amount of 1%, using single-factor experimentation and orthogonal tests to ensure the optimal conditions for E2 degradation [22]. *Rhodococcus equi* DSSKP-R-001 isolated from soil near a long-term estrogen-contaminated contraceptive plant in Beijing degraded 30 mg/L of E2 in 96 h at 30 °C and 120 rpm, using an inoculum of 3% by volume (OD600 = 1) [25]. *Rhodococcus* sp. JX-2 isolated from activated sludge degraded 90% of 30 mg/L of E2 in 7 days at pH 7.0 and 30 °C [21]. Kurisu and coworkers reported the degradation of E2 by *Rhodococcus* sp. and *Sphingomonas* sp., which were able to degrade 85% and 65% of 200 mg/L of E2 in 24 h, respectively, incubated at 25 °C with shaking at 250 rpm, using inocula adjusted by protein concentrations of 8 mg/L [15]. *Novosphingobium* sp. ES2-1 isolated from the activated sludge in a domestic sewage treatment plant could degrade 97.1% E2 (20 mg/L) in 7 days (30 °C, 150 rpm), using an inoculum of 5% (*v*/*v*) cell suspension with final optical density at 600 nm (OD600 nm) of 1.0 [26]. Fernández and coworkers reported the degradation of E2 (5 mg/L) by bacteria isolated from deep sea sediments, and faster E2 degradation occurred in 14 days by an unidentified bacterial strain incubated on a rotary shaker at 26 °C [16]. *Stenotrophomonas maltophilia* SJTL3 isolated from wastewater was found to be able to degrade 95% of 1 mg/L E2 in 6 days and 90% of 10 mg/L in 14 days, using an inoculum with the initial OD600 of 0.05, cultured at 30 °C in a shaker (200 rpm) [17]. *Bacillus* sp. E2Y4 from activated sludge degraded 1 mg/L of E2 in 4 days, using an OD600 of 0.1, incubated on a rotary shaker at 150 rpm at 30 °C [18]. E2 was nearly complete biodegraded within 72 h by *Acinetobacter* sp. LY1 at initial concentrations from 1 to 60 mg/L, incubated at 28 °C on a rotary shaker (100 rpm) [19]. Roh and Chu reported that, while *Sphingomonas* sp. ARI-1 lost its ability for estrogen after grown in the nutrient-rich medium without E2, *Sphingomonas* sp. KC8 was able to degrade 62% of a supplied amount of E2 of 2.5 mg/L in 5 days after grown on agar plates without E2, incubated at room temperature (23 °C) at 180 rpm on a shaker [27]. The observed rate was much slower than the one obtained for the cells pre-grown in medium containing E2, in which E2 was rapidly degraded to near zero within 24 h [27]. On the other hand, in the present study, *Rhodococcus* sp. ED55 revealed to be able to rapidly degrade the total amount of E2 supplied (5 mg/L) after grown on nutrient agar plates without exposure to E2 and no loss of activity was observed even after several replications were performed before the degradation experiments.

### 2.3. Biodegradation of E2 in Wastewater

The ability of the *Rhodococcus* sp. ED55 to degrade E2 in real wastewater (WW) was evaluated using samples collected from the primary clarifier from a municipal WWTP. Overall, the results displayed in Figure 3 demonstrated that in the experiment in which the wastewater was used as collected without any further treatment, total removal of 5 mg/L E2 was observed in 1.25 h when the wastewater was inoculated with strain ED55, while in the non-inoculated wastewater, E2 removal was 73% in 24 h. In the experiment performed with autoclaved wastewater, the degradation of E2 in bioaugmented flasks occurred in 2 h (Figure 3). The E2 degradation rate constants of these experiments were well fitted to the first-order kinetics (the value of R^2^ = 0.95–0.98). Degradation in non-autoclaved wastewater was slightly faster, suggesting the contribution of the bacterial community from the wastewater to the degradation. In the control of autoclaved wastewater without inoculation, the E2 concentration remained constant along the experimental period (data not shown), indicating no abiotic degradation.

In the present study, the bioaugmentation of real wastewater with *Rhodococcus* sp. ED55 significantly improved the degradation E2 from more than 24 h to less than 2 h. There are only a few studies reporting the bioaugmentation with bacterial degrading strains to improve the removal of E2 from wastewater or from natural samples. Iasur-Kruh and coworkers evaluated the incorporation of *Novosphingobium* sp. EDB-LI1 into a multiple-species mature wetland biofilm and verified the successful bioaugmentation as well as E2 removal from minimal medium containing 50 or 100 mg/L E2, using 1 g of biofilm incubated at 30 °C [28]. The biodegradation of E2 by *S. maltophilia* SJTL3 was investigated in autoclaved soil samples with the moisture adjusted to 25%. The E2 removal efficiency in the inoculated soil (cell inoculum 5% *v*/*w*) reached 99.7% (1 μg/g of E2) and 85.3% (10 μg/g of E2), respectively, after 15 days at 30 °C, which was significantly higher than those in the control groups without inoculation (15.0% and 14.3%) [17]. The E2 degradation efficiency of strain *Novosphingobium* sp. E2S was evaluated in cow manure collected from a dairy farm in Nanjing, China. The moisture content was adjusted to 50% and manure was inoculated with 5% (*v*/*w*) cell suspension and cultured at 30 °C, supplemented with 50 mg/Kg E2. The residual concentrations of E2 after seven days were 16.8 mg/kg, which was significantly lower than the residual concentration on non-inoculated control (42.1 mg/kg) [20]. *Rhodococcus* sp. JX-2, immobilized in 4% sodium alginate (1:1 bacteria/sodium alginate ratio and 5% CaCl_2_⋅2H_2_O, 6 h crosslinking time) was used for E2 degradation in sewage and livestock manure. The immobilized bacteria pellets (50 g) were introduced into 100 g freeze-dried cow dung samples (60–70% water content) from a dairy farm in Nanjing, China. The immobilized bacteria could remove more than 81% of the E2 when incubated at 30 °C for 7 days. The immobilized bacteria pellets (100 g) were also introduced into 1 L of natural sewage samples (Nanjing, China) and incubated at 30 °C for 7 days on a rotary shaker at 150 rpm. Removal efficiencies between 64.4% and almost total for E2 concentrations in the range of ng/L were observed [21]. In our study, the degradation efficiency of *Rhodococcus* sp. ED55 was not negatively influenced by the presence of various wastewater organics or by the microbial community present in the wastewater. This reveals its potential to be used in bioaugmentation for the degradation of E2 in real wastewater, what would be of great value, since E2 enters waterways via effluents of municipal WWTPs [29].

### 2.4. Identification of Metabolites Produced by E2 Biodegradation

The efficiency of *Rhodococcus* sp. ED55 in removing E2 in real wastewater was combined with the determination and identification of metabolites through using liquid chromatography, coupled with quadrupole-time-of-flight mass spectrometry (LC-QTOF-MS). Several improvements in MS techniques have been carried out to date in the structural elucidation of biotic and abiotic metabolites of steroid hormones in the environment [30].

All the samples collected during the experiments, performed in both unsterile and sterile real wastewater supplemented with 5 mg/L of E2 without the addition of the bacterial strain and bioaugmented with *Rhodococcus* sp. ED55, were analyzed for metabolites by UPLC/ESI-QTOF-MS-MS.

In the first step of processing, the suspect screening, based on a list of metabolites already known from the literature in which E2 biodegradation was investigated, allowed the identification of the formation of four metabolites through the combination of the accurate mass, isotopic cluster of molecular ion and MS/MS fragmentation pattern. The analytical protocol employed for non-target screening also allowed the detection of a list of molecular ions subsequently reduced by the intersection of replicates and removal of peaks also detected in blank samples. By the careful processing of the detected peaks, 13 molecular ions were selected and confidently associated with metabolites. In Table 1, the calculated and measured masses of the 17 identified metabolites, as well as the ionization acquisition mode, the MS/MS fragments and the predicted formula are listed.

For 7 out of 17 detected metabolites (metabolites 1–8), a chemical structure was proposed, allowing the prediction of a biotransformation pathway of E2 by the bacterial strain ED55 (Figure 4).

As already mentioned in several works, the dehydrogenation of E2′s ring D at C-17 generates the metabolite 1 (identified as E1, estrone) with two fewer hydrogen atoms with respect to E2 and the formation of a ketone group [15,22,31]. For metabolite 2, a mass increase of 16 Da in relation to E1 reveals a subsequent addition of one oxygen atom, explained by the hydroxylation on E1′s ring A (identified as 4-OH-E1). Alternatively, the cleavage of E1′s ring D followed by di-oxidation can lead to the formation of metabolite 3, reduced in the subsequent steps of the biotransformation pathway to the metabolite 4 [32,33].

Kurisu et al., proposed the hydroxylation of E2 at C-4 without first being degraded to E1, thus explaining the formation of the metabolite 5, identified as 4-OH-E2 (*m*/*z* 288.1725) [15]. Otherwise, the dehydrogenation of E2′s ring A can generate the formation of metabolite 6 (*m*/*z* 276.2089).

The production of metabolite 7 (*m*/*z* 299.1521), another estrogen degradation metabolite (namely pyridinestrone acid), was documented in previous work and its formation mechanism was ascribed to the condensation of phenol ring-cleavage products with ammonium ions [26,31,34].

As for the other 10 detected ions (metabolites 8–17), mainly nitrogen compounds, a predicted formula was assigned using the spectral information, but insufficient data existed to propose possible structures (identification confidence of level 4). However, the formula provides some information and is worth presenting, as it can be traced in future studies.

The obtained list of metabolites was then processed in SciexOS software 1.2 (Sciex, Framingham, MA, USA) for detecting significant trends. All the profiles achieved by plotting the peak area of the selected metabolites along time are shown in Figure 5. Particularly, two different types of profiles, namely a bell-shape trend or a constant increase along reaction time, were obtained.

The present work, as well as the work by Lee and Liu [32], Chen et al. [31], Li et al. [26], suggested that the initial transformation of E2 after exposition to sewage microorganisms occurred in ring D, producing the estrogen E1 (estrone). As shown in the time profile in Figure 5, both E1 (metabolite 1) and 4-OH-E1 (metabolite 2) are produced from the metabolism of the ED55 and other bacteria from WW being detected in unsterile and sterile WW samples when inoculated with the bacterial strain. Another estrogen-like chemical, namely pyridinestrone acid (metabolite 7), already documented in previous reports [26,31,34] was detected only in WW unsterile samples both in the presence and absence of ED55, suggesting that it is produced by ED55 and other bacteria from WW. The same profile was observed for metabolites 3, 14 and 15 with a constant increase in intensity along the reaction time. Differently, metabolites 13, 16 and 17 are clearly produced from the biodegradation of E2 by *Rhodococcus* sp. ED55, being detected only in WW sterile supplemented with bacterial strain. In fact, since the wastewater is autoclaved, it is most likely that the E2 degradation is due to the presence of the bacterial ED55. Moreover, it is worth noting that for the metabolites E1, 4-OH-E1 and 4-OH-E2 (metabolite 1, 2 and 5, respectively), even in the presence of the bacterial community naturally present in the wastewater (WW unsterile), the bioaugmentation with ED55 determines a faster and complete removal of the produced metabolites.

### 2.5. Toxicity Tests

The risk of producing metabolites with higher toxicity than the parent compound was evaluated by monitoring changes in whole sample toxicity. The results obtained for the *Allivibrio fischeri* bioluminescence inhibition revealed a decrease in whole sample toxicity at the end of the experiments, in sterile and unsterile wastewater (Table 2). In the assay with sterile wastewater, a reduction in toxicity was only observed in the assays bioaugmented with *Rhodococcus* sp. ED55 and not in the control assay (Table 2). In the experiments performed in MM, no inhibition of bioluminescence was observed (data not shown).

In relation to the toxicity experiments using *L. sativa* in sterile wastewater, a 30% reduction in the inhibition of germination was observed, whereas an 84% reduction in the inhibition of root growth and total removal of the inhibition of shoot growth was observed in the assays bioaugmented with *Rhodococcus* sp. ED55. In the control assays without the inoculation of strain ED55, the samples did not present a reduction in the inhibition effect over *L. sativa* (Table 3). In the unsterile wastewater, a 34% reduction in the inhibition of germination and total removal of the negative effect on the growth in all samples after the biodegradation experiments was observed (Table 3). In MM, the toxic effect of E2 was only observed in the germination and a 44% reduction in the inhibition after the biodegradation in relation to the beginning was noticed (Table 3).

### 2.6. Estrogenic Activity

The estrogenic activity can be described as an effect caused by endocrine disrupting chemicals that interfere with or damage the organism’s endocrine system. Total removal of estrogenic activity was observed in unsterile wastewater, both in the bioaugmented and non-bioaugmented assays (Table 4). In sterile wastewater and in MM, the degradation of 5 mg/L of E2 by *Rhodococcus* sp. ED55 resulted in a significant decrease in estrogenic activity in the assays inoculated with the bacterial strain (Table 4), indicating that the degradation products had lower estrogenic activity. A reduction in the whole sample estrogenic activity after E2 biodegradation experiments was also previously reported for *Bacillus* spp. strains and *Nitrosomonas europaea* [18,35]. The removal of toxicity and the estrogenic activity was shown after biodegradation by *Rhodococcus* sp. ED55.

## 3. Materials and Methods

### 3.1. Chemicals

β-Estradiol (E2) ≥98% purity and trifluoroacetic acid 99% were obtained from Sigma-Aldrich Chemie (Steinheim, Germany). All the solvents used for analytical determinations and for preparing standard solutions, e.g., acetonitrile, methanol and formic acid, were purchased from Merck (Darmstadt, Germany). HPLC grade solvents were filtered with 0.45 μm Glass microfiber filters (Whatman™, Maidstone, UK). A Milli-Q Gradient A-10 (Millipore, Burlington, MA, USA) system was used for delivering ultrapure water (18.2 MΩcm, organic carbon ≤ 4 μg/L). Minimal salts medium (MM) [36] was prepared with analytical grade chemicals (Sigma-Aldrich Chemie, Steinheim, Germany; Merck, Darmstadt, Germany). Sodium acetate was purchased from Merck (Darmstadt, Germany).

### 3.2. E2 Degrading Strain

The E2 degrading strain named ED55 was previously isolated after the enrichment of sediments of a discharge point of a WWTP in Coloane, Macau [24]. In order to obtain the complete sequence of the 16S rRNA gene, genomic DNA extraction and further amplification by polymerase chain reaction (PCR) were performed as described elsewhere [37]. Sequencing was performed at Eurofins genomics (Leipzig, Germany), using universal bacterial 16S rRNA primers (f27, f518, r800 and r1492). To determine the phylogenetic affiliation, similarity searches were conducted using the EzBioCloud [38,39]. Phylogenetic analysis was performed to accurately determine the taxonomic position of the E2 degrading strain. For this, the 16S rRNA gene sequence of strain ED55 was aligned with the reference sequences available in the GenBank/EMBL/DDBJ database. The phylogenetic tree was constructed with the MEGA software (version X) [40] using the neighbor-joining method (Kimura two-parameter distance optimized criteria). The 16S rRNA gene sequence of strain ED55 was submitted to the GenBank database under the accession number MZ422537.

### 3.3. Biodegradation of E2 in Mineral Salts Medium (MM)

To investigate E2 biodegradation by strain ED55, cells previously grown on nutrient agar (NA) plates were inoculated to an OD450 of ca. 0.05 into MM supplemented with E2 at a concentration of 5 mg/L. This concentration was chosen, although being higher than typically present in the environment, to be able to follow E2 degradation and detect metabolites and it was in the same order of magnitude as used in similar studies [16]. Experiments were performed using E2 as a single substrate and with the addition of a supplementary carbon source, sodium acetate, supplied at 5.9 mM. All the cultures were incubated at room temperature (30 °C) on a rotary shaker (130 rpm). These conditions are in line with the optimal conditions for the bacterial strain and are similar to previous studies for the degradation of estrogens by *Rhodococcus* bacteria [21,22] Experiments were performed in triplicate under sterile conditions protected from light. Control assays consisted of sealed flasks containing MM supplemented E2 as a single substrate or with the addition of acetate, without inoculation or inoculated with autoclaved strain ED55 cells. A control for cell growth was established with the same concentration of acetate without E2 addition. Samples were taken each 15 min during the first two hours of the experiment and then each hour over ten hours to assess growth and E2 degradation. The final sampling point was taken after 24 h. The purity of the cultures was evaluated through plating on NA plates.

### 3.4. Biodegradation of E2 in Wastewater

The biodegradation of E2 by strain ED55 was tested in wastewater collected from the primary clarifier from the wastewater treatment plant of Parada (Maia; greater Porto area, Portugal). Two different experiments were performed. In one case, the wastewater was used as collected, without any further treatment; in another experiment, the wastewater was sterilized by autoclaving. Wastewater was supplemented with E2 at 5 mg/L; this concentration was chosen in order to have a concentration high enough to be able to detect the produced metabolites.

The cells of strain ED55 were inoculated to an OD450 of ca. 0.1. Control assays consisted of sealed flasks containing wastewater supplemented with E2, as a single substrate without inoculation. A control for metabolites formation was established in the same conditions without E2 addition. Samples were taken each 15 min during the first two hours of the experiment and then each hour during 24 h; sampling points at 48 h and 72 h were also taken to assess E2 degradation as well as the formation of intermediate metabolites.

### 3.5. Analytical Methods and Data Processing

The residual concentration of spiked E2 along time was measured by high-performance liquid chromatography (HPLC). The HPLC analyses were performed on a Waters e2695 (Waters, MI, USA), coupled with a fluorescence detector at an excitation wavelength of 230 nm and an emission wavelength of 310 nm. Chromatographic separations were performed using a reversed phase 250-4 HPLC-Cartridge LiChrospher 100 RP-18 column (Merck, Darmstadt, Germany), which operated in isocratic mode at room temperature. Acetonitrile/water acidified to pH 2 with trifluoroacetic acid (60:40, *v*/*v*) was used as the mobile phase, with a flow rate of 1 mL/min. The injection volume was 20 µL.

The detection of metabolites in the wastewater samples was carried out with an Ultimate 3000 UPLC System (Thermo Fisher Scientific, Waltham, MA, USA) coupled with a HRMS TripleTOF^®^ 5600 System (Sciex, Framingham, MA, USA). MS analyses were performed in electrospray (ESI) positive ion mode with an acquisition method based on both full-scan survey TOF-MS and IDA (information dependent acquisition) experiments, in a scanned mass range from 65 to 1000 *m*/*z*. The chromatographic separation of the analytes was obtained using a Water BEH C18 column (250 × 2.1 mm, 1.7 μm) operating at a flow of 0.200 mL/min with a binary gradient consisting of H_2_O, 0.1% HCOOH (A) and ACN, 0.1% HCOOH (B). The gradient was optimized as follows: 0–1 min 2% B, 2–16.5 min 100% B, 16.5–24.5 min 100%B, 24.5–25.0 min 2%B, 25.0–30.0 min 2%B. The injection volume was set at 20 μL of the sample.

The collected data files were processed qualitatively using AB Sciex software (Sciex, Framingham, MA, USA), namely SciexOS, PeakView and MasterView. In the first step, a suspect screening was performed based on a list of metabolites already known from the literature. Such an approach screened for the presence of metabolites, combining exact MS error, isotope ratio difference and MS/MS fragments measured in the IDA scans, providing a tentative identification of metabolites with the confidence of level 2 [41]. To extend the search for metabolites beyond the literature, extensive screening was performed using the Metabolitepilot™ software (Sciex, Framingham, MA, USA). The approach used by this software is reported elsewhere [42] and it is principally based on biotransformation pathways, such as desaturation, demethylation, decarboxylation, oxidation and hydrogenation, which allows for the prediction of potential cleavage metabolites starting from the structure of the parent compound. Finally, a non-target screening was carried out using a data workflow in SCIEX OS software 1.2 (Sciex, Framingham, MA, USA). The resulting data from the peak finding include a list of ions with a corresponding chromatographic peak and retention time, TOF MS spectrum and corresponding Formula Finder result, and lastly, the acquired MS/MS spectrum. The list of ions was then processed by SciexOS software 1.2 (Sciex, Framingham, MA, USA) for detecting significant trends as well as for identification of unknown compounds, employing both the formula finder based on isotopic patterns and high-resolution MS-MS data to attempt a candidate structure for the identified metabolites.

### 3.6. Toxicity Tests

For the toxicity assays, samples were collected at the beginning and end of the biodegradation experiments. The biomass was removed by centrifugation at 14,000 rpm for 10 min at 4 °C and kept at −20 °C until the moment of analysis. The pH was not adjusted, and the samples were not diluted. Samples were analyzed without dilution in order to assess the evolution of whole sample toxicity and to compare the resulted final toxicity with the control experiments. All toxicity assays were performed in quadruplicate. The *Allivibrio fischreri* luminescence test was performed according to UNI EN ISO 11348-2-2007 (International Organization for Standardization, 2007), following the Luminescent Bacteria Test LCK 480 manufacturer instructions (Hach Lang GmbH, Dusseldorf, Germany). This method is based on the percentage of inhibition of light emitted by the bioluminescent bacterium *Allivibrio fischeri* upon contact during 30 min with the sample. The acute bioassay with *L. sativa* evaluated the potential toxicity considering the inhibition of seed germination and root and shoot elongation according to OECD Guideline 208. Briefly, the seeds were surface sterilized with bleach solution (5% commercial bleach) for 15 min and washed 3 times with sterile water. A total number of 10 seeds were placed in Petri dishes containing a Whatman No. 2 filter paper and exposed to 3 mL of samples; the dishes were sealed with Parafilm in an incubator at 25 °C in the dark. The experiments were performed for 7 days. After that period, the seedlings were visually checked in order to count the germinated seeds and measure root and shoot growth in the germinated seeds. The inhibition normalized on negative control data was expressed as a percentage of effect. Seedling emergence of 80% was observed in the control performed with distilled water, proving seed viability.

### 3.7. Assay of Estrogenic Activity

Estrogenic activities of the samples collected at the beginning and end of the biodegradation experiment were determined using the standardized yeast estrogenic (S-YES^MD^, *Saccharomyces*-yeast estrogen screen) assay with the recombinant strain of *Saccharomyces cerevisiae*, producing β-galactosidase in response to estrogen exposure, which was purchased from New diagnostics GmbH (Freising, Germany). All the samples were determined in triplicate. E2 was used as the positive control; serial dilutions of natural estrogen were measured in the range of 5–400 ng/L to obtain the whole dose–response curve of estradiol. The detection was performed photometrically after the reaction of the reporter enzyme with the chromogenic substrate chlorophenol red-β-D-galactopyranoside at 580 nm. Water was used as the blank control. The estrogenic activity of a sample measured by the YES assay was expressed as an estradiol equivalent concentration (EEQ). The limit of detection for the S-YES^MD^ kit was 3.8 ng/L, according to manufacturer instructions.

### 3.8. Statistical Analysis

Data were analyzed using one-way analysis of variance (ANOVA; IBM SPSS Statistics 28.0, Chicago, IL, USA), and means compared using Tukey’s HSD test (*p* < 0.05).

## 4. Conclusions

The bacterial strain *Rhodococcus* sp. ED55 was able to degrade E2, an endocrine disrupting chemical, and one of the substances on the watch list of the European Water Framework Directive (Decision 2018/840; Directive 2013/39/EU), at high degradation rates. The bacterial strain was also able to degrade E2 when used to bioaugment wastewater from a municipal WWTP. Moreover, the elimination of the toxicity and decrease in the estrogenic activity of treated effluent is a very important feature. This work shows the potential of the use of this strain for environmental remediation processes.

## Figures and Tables

**Figure 1 ijms-23-06181-f001:**
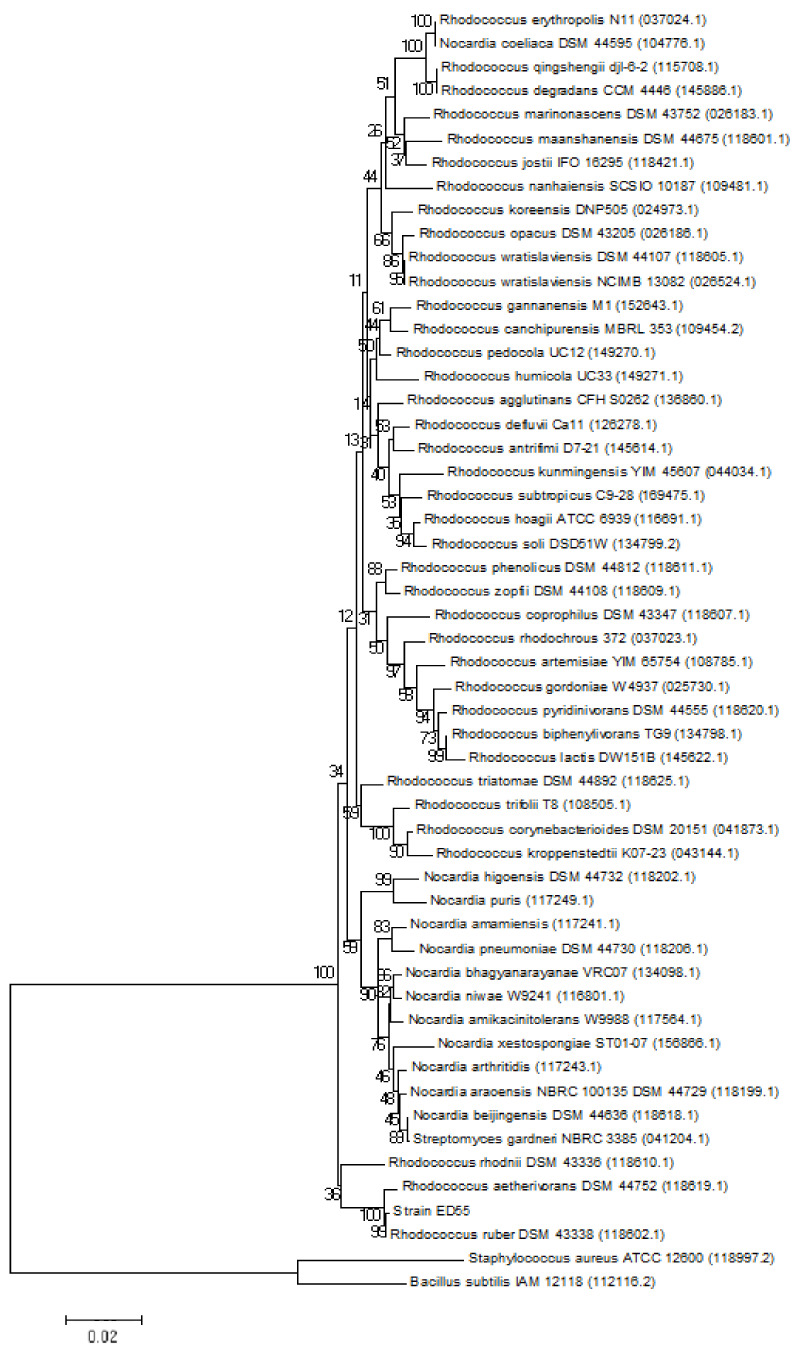
Neighbor-joining phylogenetic tree based on 16S rRNA gene sequences, showing the nearest neighbors of strain ED55. *Bacillus subtilis* and *Staphylococcus aureus* type strains were used as the outgroup. GenBank accession numbers are given in parentheses. The bar represents 0.02 substitutions per site.

**Figure 2 ijms-23-06181-f002:**
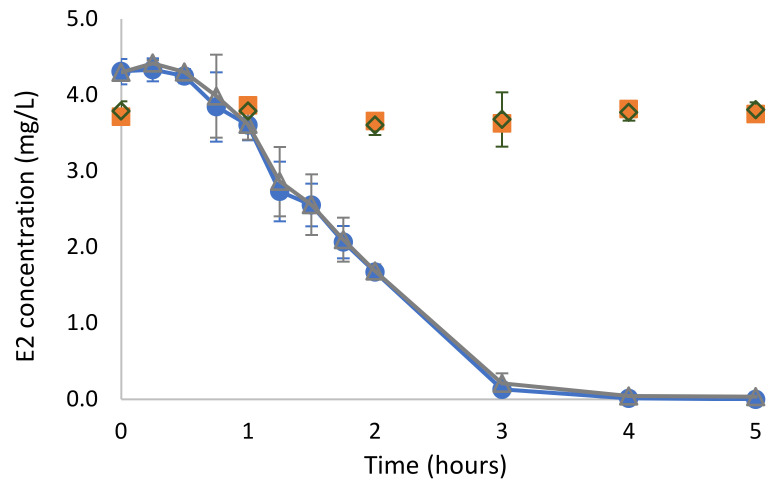
Biodegradation of E2 (5 mg/L) by *Rhodococcus* sp. ED 55 in the synthetic mineral medium as sole carbon source (●), with supplementation of sodium acetate (Δ), adsorption control (◊) and abiotic control (▪). The experiment was conducted in triplicate.

**Figure 3 ijms-23-06181-f003:**
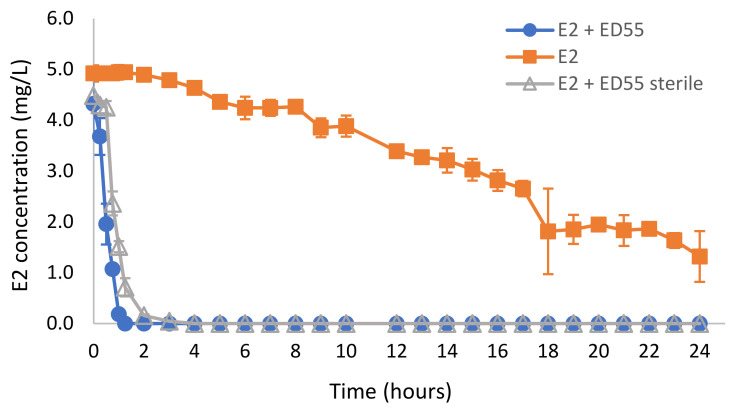
Removal of E2 in municipal wastewater: non-autoclaved wastewater without the addition of the bacterial strain (▪) and bioaugmented with *Rhodococcus* sp. ED55 (●); and autoclaved wastewater bioaugmented with *Rhodococcus* sp. ED55 (**Δ**). Error bars represent the standard deviation achieved carrying out the experiment in triplicate.

**Figure 4 ijms-23-06181-f004:**
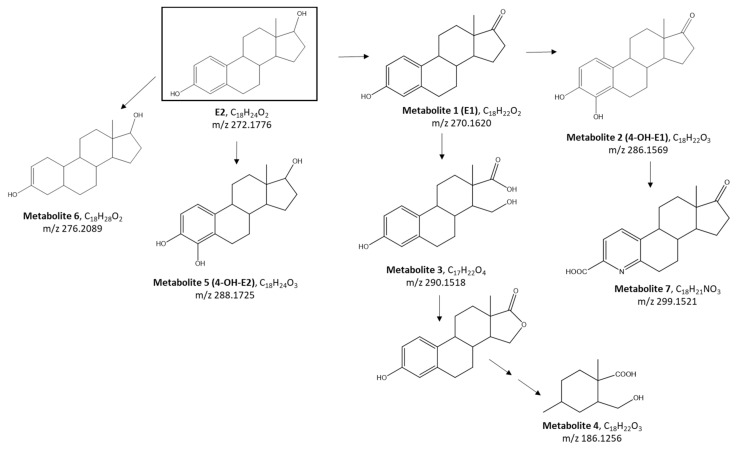
Proposed biodegradation pathway of E2, obtained based on the metabolites identified by the suspect and non-target screening.

**Figure 5 ijms-23-06181-f005:**
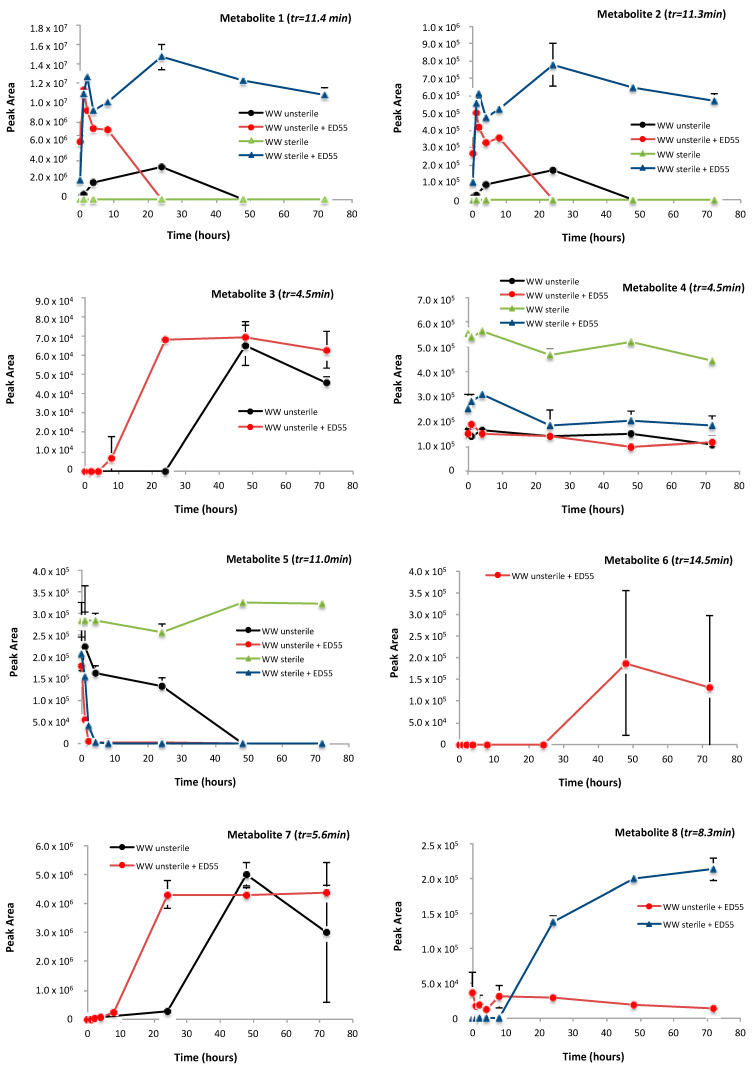
Time profiles of detected metabolites under investigated treatments, namely WW unsterile and sterile, both supplemented with E2 in the presence of bacterial strain ED55 and without the bacterial strain.

**Table 1 ijms-23-06181-t001:** E2 metabolites detected by UPLC/ESI-QTOF-MS IDA.

Metabolite	Ionization Mode	Calculated *m*/*z*	Measured *m*/*z*	Error ppm	MS/MS Products	Predicted Formula
1	ESI (-)	269.1547	269.1546	−0.5	145.0659, 183.0891, 159.0819, 253.1238	C_18_H_22_O_2_
2	ESI (-)	285.1496	285.1495	−0.5	108.0214, 161.0597, 225.1273, 267.1384	C_18_H_22_O_3_
3	ESI (-)	289.1445	289.1448	0.8	121.0665, 149.0975, 219.1391, 245.1538	C_17_H_22_O_4_
4	ESI (-)	185.1183	185.1186	1.6	79.9556, 121.0679, 139.1107, 166.9829	C_10_H_18_O_3_
5	ESI (-)	287.1652	287.1643	−3.3	108.0215, 183.0808, 259.1541, 269.1542	C_18_H_24_O_3_
6	ESI (-)	275.2016	275.2008	−3.3	159.0774, 177.1660, 203.1895, 231.2112	C_18_H_28_O_2_
7	ESI (+)	300.1592	300.1594	−0.6	145.1010, 194.0971, 236.1442, 254.1538	C_18_H_21_NO_3_
8	ESI (+)	559.2524	559.2538	−2.5	136.0751, 283.1435, 412.1846, 531.2573	C_30_H_38_O_10_
9	ESI (-)	329.2483	329.2486	−0.9	163.0727, 231.2079, 251.1948, 285.2594	C_22_H_34_O_2_
10	ESI (-)	374.2446	374.2449	−0.9	232.1093, 290.1877	C_21_H_33_N_3_O_3_
11	ESI (-)	357.2051	357.2044	1.7	121.0650, 186.9527, 235.1360, 274.9421	C_18_H_26_N_6_O_2_
12	ESI (-)	162.0784	162.0785	−0.4	90.0098, 100.9240, 117.0264, 146.0466	C_7_H_9_N_5_
13	ESI (-)	150.0421	150.0421	0	80.0294, 90.0085, 108.0194, 133.0147	C_5_H_5_N_5_O
14	ESI (+)	329.1571	329.1568	0.9	75.0256, 117.0733, 243.2328, 287.2228	C_12_H_20_N_6_O_5_
15	ESI (+)	658.2180	658.2184	0.6	171.0652, 310.0744, 456.1352, 512.1588	C_38_H_31_N_3_O_8_
16	ESI (+)	113.0341	113.0345	−4.2	70.0283, 96.0072	C_4_H_4_N_2_O_2_
17	ESI (+)	364.1134	364.1139	−1.4	152.0559, 187.0503, 230.0555, 346.1021	C_16_H_17_N_3_O_7_

**Table 2 ijms-23-06181-t002:** Acute toxicity tests performed with *Allivibrio fischeri* (% bioluminescence inhibition).

	WW Sterile	WW Unsterile
Before	55.5 ± 1.0 ^a^	50.2 ± 2.0 ^b^
After (without ED55)	49.2 ± 2.3 ^b^	12.6 ± 0.7 ^c^
After (with ED55)	2.3 ± 3.6 ^d^	0.0 ± 0.0 ^d^

Results are expressed as mean ± SD. Means with different letters differed significantly according to Tukey’s HSD test at *p* < 0.001.

**Table 3 ijms-23-06181-t003:** Acute toxicity tests performed with *Lactuca sativa*.

	Germination (% Inhibition)			Root Growth (% Inhibition)			Shoot Growth (% Inhibition)		
	WW Sterile	WW Unsterile	MM	WW Sterile	WW Unsterile	MM	WW Sterile	WW Unsterile	MM
Before	57.1 ± 7.1 ^d^	52.9 ± 8.7 ^cd^	28.6 ± 4.3 ^ab^	35.6 ± 4.6 ^c^	19.0 ± 2.0 ^b^	0.0 ± 0.0 ^a^	5.32 ± 0.9 ^ab^	12.4 ± 6.5 ^c^	4.37 ± 0.6 ^a^
After (without ED55)	56.6 ± 8.3 ^cd^	33.3 ± 2.0 ^b^	30.2 ± 4.6 ^ab^	39.7 ± 3.8 ^c^	0.0 ± 0.0 ^a^	0.0 ± 0.0 ^a^	5.22 ± 1.3 ^ab^	0.0 ± 0.0 ^a^	0.0 ± 0.0 ^a^
After (with ED55)	39.7 ± 2.8 ^bc^	34.9 ± 6.6 ^b^	15.9 ± 6.0 ^a^	5.63 ± 4.8 ^a^	0.0 ± 0.0 ^a^	5.0 ± 1.6 ^a^	0.0 ± 0.0 ^a^	0.0 ± 0.0 ^a^	2.34 ± 0.3 ^a^

Results are expressed as mean ± SD. Means with different letters in the same parameter differed significantly according to Tukey’s HSD test at *p* < 0.001.

**Table 4 ijms-23-06181-t004:** Estrogenic activity measured using S-YES^MD^ kit expressed in E2 equivalents (EEQ) in ng/L.

	WW Sterile	WW Unsterile	MM
Before	>400 ^a^	>400 ^a^	352.2 ± 14.1 ^d^
After (without ED55)	>400 ^a^	<LOD ^c^	>400 ^a^
After (with ED55)	126.3 ± 13.0 ^b^	<LOD ^c^	124.9 ± 19.3 ^b^

Results are expressed as mean ± SD; LOD limit of detection. Means with different letters differed significantly according to Tukey’s HSD test at *p* < 0.001.

## Data Availability

The 16S rRNA gene sequence of *Rhodococcus* sp. ED55 was deposited in the GenBank database [43].

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
