# Peer review of "Biodegradation and Metabolic Pathway of 17β-Estradiol by Rhodococcus sp. ED55"

_ijms, 2022, doi:10.3390/ijms23116181_

Round 1

Reviewer 1 Report

The manuscript (Ms.) is a well come contribution to the knowledge of 17β-estradiol biodegradation of an actinobacterial strain (Rhodococcus sp. ED55) in synthetic medium and real wastewater conditions. Along the reading of the Ms. I had some remarks/questions and suggestions.

General remarks:

  • Uniform the concentration units writing: mg(ng)/L (sometimes is l others L; sometimes mg/L others mgL-1). Also, uniform liter (I or L; preferable the latter);
  • Substitute V. fischeri per Aliivibrio fischeri.Leave a space between a number and its unit (e.g. 3.5 h and not 3.5h)
  • All species names must be in italic
  • The results decimals are points (.), not comma (,) – confirm all numbers along the text and in tables (2, 3, and 4)
  • Statistical analyses should be presented.

Introduction:

  • Lines 47-48 “Concentrations of E2 and EE2 in surface water are typically below 50 ng/l and 10 ng/l respectively” – could you support with references?
  • Line 63 – “Rhodococcus sp. sp. JX-2 [19]” – remove one of the two sp.

M&M:

  • Line 319, “was submitted to the GenBank database under accession number MZ422537.” – please confirm if the accession number is correct
  • Point 4.3. – explain the E2 concentration (5 mg L-1 is considerable higher than 50 ng/L and 10 ng/L. Why bacterial growth at 30 ºC? Line 333 seems a repletion of line 331. In these lines and lines 344/345 instead of “regular intervals” stated the intervals: e.g. every X hours/days.
  • Lines 384/385 “For toxicity assays, samples were collected at the beginning and end of the biodegradation experiment for toxicity assays.” - correct to “For toxicity assays, samples were collected at the beginning and end of the biodegradation experiment.”
  • Lines 385/386 “Samples were analysed without dilution in order to assess the evolution of whole sample toxicity and to compare the resulted final toxicity with the control experiments.” – explained better: was the pH adjusted? It is not clear the absence of dilutions.
  • Point 4.6 – further explains of the germination test is needed: number of seeds per treatment, sterilization of seeds, seed viability;
  • Point 4.7 – the Limit of Detection of the test must be stated, as well the wavelength used.
  • Line 400: put the name of the species “Saccharomyces cerevisiae” in italic; also, substitute “b-galactosidase” per “β-galactosidase”

Results:

- revise lines 82/83 – The observation under a light microscope does not reveal that a bacterium is gram-positive;

- In Fig. 1 - mark/show, in the figure, the strain;

- line 97 – “reaction rate of degradation”, substitute per “rate of degradation”;

- Fig. 2 – all the tested conditions should be represented in the graphic;

- Fig. 3 – grey triangles are “bioaugmented autoclaved wastewater” – bioaugmented with Rhodococcus sp. ED55? Correct the triangles: in graphic the triangles are in full grey, while in the legend are not.

- between lines 150-164 the results are discussed; for simplicity and coherency, considerer joint the results and discussion sections;

- the quality of figure 5 should be improved (has overlapping information);

- Table 2 – values units? Standard-error or standard-deviation? Statistical analyses?

- “not sterile” is “unsterile”

- Table 3 – Standard-error or standard-deviation? Statistical analyses? The germination is not normal: germination without root growth, or germination without root growth, but with shoot growth. How the seed sterilization was done? Seed germination is highly affected by many parameters such as salinity, pH.

- Table 4 – explain, in the caption, the acronym LOD. Any replicates? Standard-error or standard-deviation? Statistical analyses?

Discussion

  • Discussion of E2 degradation percentages should be taken into account not only its initial concentration and the incubation media, but also the incubation conditions (temperature, agitation, and reactor type, e.g. batch/fed-batch/continuous) and bacteria taxonomy and metabolism. Without these type of information it is difficult to follow the discussion

Author Response

We are grateful for the valuables comments of the reviewer, which improved the manuscript. Please see the attachment for a point-by-point response.

Reviewer 2 Report

Well written article, clearly stated and of high importance as endocrine disruptors, especially those that bind to estrogen receptors are of concern due to the massive plastic pollution we are facing.

I would like to see in the introduction a few words regarding Bisphenol A (an plastic additive) that has similar activity (as I can see, it's mentioned in 2 citations but not in text).
4.5. Analytical methods and Data processing - Clarification: HPLC reading was performed at ? nm? 

Very well written paper, congrats!

Author Response

The authors appreciate the comments and suggestions of the reviewer, which helped improved the quality of the manuscript. Please see the attachment for a point-by-point response.

Round 2

Reviewer 1 Report

I very much appreciate the changes you have made to your manuscript. Excellent job.